# Correlation between Muscular Activity and Vehicle Motion during Double Lane Change Driving

**DOI:** 10.3390/s24185982

**Published:** 2024-09-15

**Authors:** Myung-Chul Jung, Seung-Min Mo

**Affiliations:** 1Department of Industrial Engineering, Ajou University, 206, World Cup-ro, Yeongtong-gu, Suwon-si 16499, Gyeonggi-do, Republic of Korea; mcjung@ajou.ac.kr; 2Department of Occupational Safety and Health Management, Osan University, 45 Cheonghak-ro, Osan-si 18119, Gyeonggi-do, Republic of Korea

**Keywords:** electromyography, vehicle motion, correlation, double lane change, muscular activity

## Abstract

The aim of this study was to compare the correlation between electromyography (EMG) activity and vehicle motion during double lane change driving. This study measured five vehicle motions: the steering wheel angle, steering wheel torque, lateral acceleration, roll angle, and yaw velocity. The EMG activity for 19 muscles and vehicle motions was applied for envelope detection. There was a significantly high positive correlation between muscles (mean correlation coefficient) for sternocleidomastoid (0.62) and biceps brachii (0.71) and vehicle motions for steering wheel angle, steering wheel torque, lateral acceleration, and yaw velocity, but a negative correlation between the muscles for middle deltoid (−0.75) and triceps brachii long head (−0.78) and these vehicle motions. The ANOVA test was used to analyze statistically significant differences in the main and interaction effects of muscle and vehicle speed. The mean absolute correlation coefficient exhibited an increasing trend with the increasing vehicle speed for the muscles (increasing rate%): upper trapezius (30.5%), pectoralis major sternal (38.7%), serratus anterior (13.3%), and biceps brachii (11.0%). The mean absolute correlation coefficient showed a decreasing trend with increasing vehicle speed for the masseter (−9.6%), sternocleidomastoid (−12.9%), middle deltoid (−5.5%), posterior deltoid (−20.0%), pectoralis major clavicular (−13.4%), and triceps brachii long head (−6.3%). The sternocleidomastoid muscle may decrease with increasing vehicle speed as the neck rotation decreases. As shoulder stabilizers, the upper trapezius, pectoralis major sternal, and serratus anterior muscles are considered to play a primary role in maintaining body balance. This study suggests that the primary muscles reflecting vehicle motions include the sternocleidomastoid, deltoid, upper trapezius, pectoralis major sternal, serratus anterior, biceps, and triceps muscles under real driving conditions.

## 1. Introduction

Automotive technology has been researching improvements in vehicle performance to ensure a stable and comfortable ride for drivers and passengers [1,2]. The performance of the vehicle was assessed across parameters, such as motion, stability, and response time, serving as foundational data for further vehicle development. Vehicle motion has been measured by using devices such as wheel tracking devices, steering wheel robots, load cells, and global positioning systems [3,4,5]. These devices evaluate vehicle motions, providing crucial mechanical data such as steering wheel angle and torque, lateral acceleration, roll angle, and yaw velocity, which in turn informs the development of vehicles [6,7,8]. However, the use of these devices has certain limitations. Installation of the devices could be time-consuming, and they primarily assess vehicle motion and response time, except for the driver’s physiological response.

The evaluation of physiological and subjective responses should be considered to ensure the driver’s needs. Considering the safety, comfort, and convenience of the driver, mechanical improvements have become increasingly important [9,10]. The vehicle ergonomic group in the UK has been evaluated as a subjective measure of various vehicles during road trials since the 1980’s. To assess drivers’ misjudgement, Schmidt et al. [11] recorded subjective, performance, and physiological measures, such as monotony rate, reaction time, electroencephalography, and electrocardiograph. During monotonous daytime driving, there was a significant linear increase in the alpha spindle rate on electroencephalography, accompanied by a significant linear decrease in heart rate. Durkin et al. [12] determined the effects of massage seats on physiological measures such as electromyography (EMG) fatigue for lumbar muscles, oxygenation, and blood flow during prolonged driving. The results showed that EMG fatigue was not significantly different, suggesting that the massage seat contributed to reduced lumbar discomfort. Farah et al. [13] discovered a significant correlation between EMG activity for the cervical erector spinae, external oblique, and vastus lateralis muscles and lateral acceleration, with a correlation coefficient exceeding 0.85. This showed that the EMG activity could be significantly different as the vehicle motion and performance changed. It is important to understand the relationship between the driver’s physiological response and vehicle motion [14,15]. Thus, EMG activity can provide useful information on muscle activity during the driving condition.

Wilder et al. [16] investigated erector spinae muscle fatigue in the lower back using the median power frequency for the EMG signal under simulated driving conditions involving a suspension of the vehicle seat and typical driving postures. There was no significant difference in muscle fatigue between suspension and posture in the short term. El Falou et al. [17] evaluated muscle fatigue, subjective discomfort, and task performance for vibrating car seats during long-duration driving. Muscle fatigue for the cervical erector spinae and external oblique muscles did not exhibit a significant difference in experimental conditions or across time, whereas subjective discomfort significantly increased. Pick and Cole [18] evaluated muscle co-contraction stiffness in the generation of steer torque while simulated double lane change. They reported that the anterior and middle deltoid, pectoralis major clavicular portion, and triceps brachii long head showed significantly strong multiple correlation coefficients of approximately 0.9 using multiple regression analysis. Balasubramanian and Adalarasu [19] analyzed muscle fatigue for the middle deltoid and upper trapezius of professional and non-professional drivers in a 15-min simulated driving condition. They also found that there was a significant difference between professional and non-professional drivers during the last minute. Gao et al. [20] analyzed the response of vehicle parameters and EMG of the tibialis anterior muscle. They found that drivers may be at risk of crashing while maintaining their normal driving posture, as certain muscles may not be activated or fully activated during very urgent situations. Previous studies have examined the relationship between vehicle motion and the driver’s physiological responses by measuring the EMG activity of muscles, such as the erector spinae, biceps brachii, and biceps femoris, which are directly relevant to driving.

Even though facial and neck muscles are irrelevant to steering wheel control, EMG activities of the masseter muscle showed significant differences with vehicle motion [21]. Zheng et al. [22] evaluated the sternocleidomastoid muscle activity of a passenger in response to a car’s lateral acceleration during slalom driving. The study revealed that passengers experience increased discomfort as the EMG signal of the sternocleidomastoid muscle increases, and conversely, they feel less discomfort as the signal decreases. The simulator driving conditions of previous studies were somewhat different from those of real driving conditions. This is because the motion of the driver is affected by changes in the vehicle speed or inertial moment in real driving conditions. The real driving condition may be influenced by muscle strain owing to the driver’s psychological tension as the vehicle speed varies. To identify the steering control process, the neuromuscular system of the driver is important [23]. It is necessary to determine the primary muscles that can reflect the vehicle motion, not only the arm and shoulder muscles but also other muscles that are irrelevant to the steering wheel and pedal.

Therefore, the primary aim of this study was to compare the correlation between EMG activity and vehicle motion during double lane change conditions. The secondary aim was to determine the muscles by the reflected vehicle motion using a correlation coefficient. Accordingly, this study hypothesizes that there is no correlation between EMG activity and vehicle motion. This study offers valuable suggestions for identifying the primary muscles that reflect vehicle motion in real driving scenarios.

## 2. Materials and Methods

### 2.1. Subject

A statistical power analysis using G*Power ver. 3.1.9.6 (G*Power, Düsseldorf, Germany) was conducted to determine the appropriate number of subjects to achieve an alpha of 0.05, a power of 0.95, and an effect size of 0.9 [24]. Five skilled right-handed male drivers with over 10 years of professional driving experience volunteered to participate in this study. The Ethics Committee of the university approved the experimental methodology, and all subjects provided written consent before taking part in this study. The experimental protocol was conducted in accordance with the Helsinki Declaration as revised in 2008. All subjects provided informed written consent to participate in this experiment. None of the subjects reported any musculoskeletal or neurological disorders in the past 12 months. All the subjects were familiarized with the overall protocol prior to the experiment. The mean (standard deviation) of their age, height, and body weight were 45.4 (4.3) years, 173.3 (5.8) cm, and 75.2 (6.9) kg, respectively.

### 2.2. Apparatus

The vehicle used in the experiment was a midsized left-handed hatchback style equipped with power steering, automatic transmission, and air conditioning systems. The internal temperature of the vehicle was maintained at a constant approximate of 22 °C to minimize sweating.

EMG signals were collected by using circular silver-silver chloride bipolar electrodes with a diameter of 10 mm and an inter-electrode distance of 20 mm, which were connected to a TeleMyo 2400T DTS telemetry system (Noraxon USA, Inc., Scottsdale, AZ, USA). The raw EMG signals were amplified with a gain of 500, noise < 1 µV, and a common mode rejection ratio of 100 and sampled at a rate of 1500 Hz with a filtering bandwidth of 10–500 Hz. All data were acquired on a laptop with a 16-bit analog-to-digital converter.

This study referenced the vehicle axis system defined by the Society of Automotive Engineers [25], as shown in Table 1. The x-axis represents the longitudinal plane that extends forward from the vehicle. The y-axis extends laterally from the right side of the vehicle, is horizontal, and is positioned at a 90° angle to the x-axis. The z-axis was vertical to the other two axes, with the positive direction pointing downward. This study measured five vehicle motions: steering wheel angle (SWA), steering wheel torque (SWT), lateral acceleration (LatAcc), roll angle (Roll), and yaw velocity (YawVel), following the ISO 7401 recommended variables [26] using a DEWE-5000-PM (Dewetron, Austria) with a sampling at a rate of 200 Hz through a 16-bit analog to digital converter. Negative values occurred when turning the left side, whereas positive values occurred when turning the right side. To synchronize the EMG and vehicle motion signals, an analog output receiver (TeleMyo 2400R G2, Noraxon USA, Inc., Scottsdale, AZ, USA) was used with a 5 millivolt input TTL signal using the trigger button. To ensure the reliability of data collection, all apparatus was calibrated according to the manual.

### 2.3. Experimental Design

This study considered three independent variables: the muscle, vehicle motion, and vehicle speed. EMG signals were recorded from 19 muscles: the anterior temporalis (AT), masseter (MA), midcervical paraspinal C4 (MCP), sternocleidomastoid (SCM), upper trapezius (UT), anterior deltoid (AD), middle deltoid (MD), posterior deltoid (PD), clavicular and sternal parts of the pectoralis major (PMC and PMS), serratus anterior (SA), biceps brachii (BB), triceps brachii long head (TL), flexor carpi radialis (FCR), extensor carpi radialis (ECR), flexor digitorum superficialis (FDS), flexor digitorum profundus (FDP), and extersor digitorum communis (EDC), and rectus abdominis (RA). Table 2 lists the 19 positions of the surface electrode on the right side.

As shown in Table 2, this study measured five vehicle motions: steering wheel angle (SWA), steering wheel torque (SWT), lateral acceleration (LatAcc), roll angle (Roll), and yaw velocity (YawVel) following the ISO 7401 recommended variables [26] using a DEWE-5000-PM (Dewetron, Austria) with a sampled at a rate of 200 Hz through a 16-bit analog to digital converter. Negative values occurred when turning the left side, whereas positive values occurred when turning the right side.

The driver was randomly assigned to drive through the course in five repetitions at low (60 km/h), normal (80 km/h), high (100 km/h), and severe (110 km/h) speeds. These speeds were chosen with 20 km/h steps higher or lower than the recommended speed of 80 km/h [27]. For the selection of the severe speed at 110 km/h, it was considered challenging to pass through comfortably at 120 km/h.

As a dependent variable, this study analyzed the Pearson correlation coefficient between EMG activity and vehicle motion signals to determine whether there was a high correlation (r > 0.6) [28].

### 2.4. Procedure

Prior to attaching the electrode, the skin overlaying the muscles was shaved and cleaned with alcohol gauze to minimize the impedance. Following this, electrode positions were marked on the skin using a waterproof pen and aligned [29,30] as shown in Table 1.

Before starting the experiment, the driver was asked to perform a practice drive on the double lane change (DLC) course. The DLC course can be more effective in identifying the vehicle motion [31,32]. This DLC course is represented in Figure 1 for the ISO 3888-1 track of double lane change recommendation [27]. The vehicle to be tested was driven through the course. In this study, a lane width in sections A (2.22 m), section B (2.40 m), and section C (2.58 m) was used according to vehicle width (1.79 m). The total length of the DLC course was 125 m.

The driver positioned both hands at the 10 and 2 o’clock positions on the steering wheel while ensuring that their lower back was close to the seatback of the driver’s car seat with the seat belt securely fastened. The driver was allowed to break for more than 10 min between the randomized experimental conditions to prepare for the next condition.

### 2.5. Signal Processing

The EMG signals and vehicle motions were extracted from 0.5 s before the initial time to 0.5 s after the final time of each DLC course, excluding steady-state periods. This study collected data from a limited number of subjects, which may lead to bias in the dataset.

For the analysis of EMG activity, all heartbeat noises present in the raw EMG signals were reduced using a pattern recognition function (Noraxon USA, Inc., Scottsdale, AZ, USA). After full wave rectification of the EMG signals, a second-order dual-pass Butterworth low-pass filter with a cutoff frequency of 2 Hz was applied for envelope detection. The vehicle motion variables were also filtered by a second-order dual-pass Butterworth low-pass filter with a cutoff of 5 Hz to reduce unwanted noise and phase shift [33]. To allow for comparisons between the EMG activity and vehicle motion, both signals were down sampled at 100 Hz. Signal processing was performed using MATLAB (version 7.0.4, MathWorks, Inc., Natick, MA, USA). Figure 2 shows example signals of EMG muscle activity and vehicle motion during double lane change driving at a speed of 100 km/h.

The normal distribution of the dataset was tested through a one-sample Kolmogorov–Smirnov test. Descriptive statistics, correlation analysis, and repeated-measures analysis of variance (ANOVA) with Tukey’s test were performed. A significance level of 0.05 was used for all statistical analyses, which were performed using SAS software 9.4 (SAS Institute, Cary, NC, USA).

## 3. Results

### 3.1. Correlation

Table 3 shows the mean correlation coefficients between muscle and vehicle motion. Generally, there was a significantly high positive correlation between muscles for SCM and BB and vehicle motions for SWA, SWT, LatAcc, and YawVel, but a negative correlation between the muscles for MD and TL and these vehicle motions. In contrast, the MD, PD, and TL muscles exhibited a significantly high positive correlation with vehicle motion for Roll, while the SCM and BB muscles showed a significantly negative correlation with roll.

### 3.2. ANOVA

The results of the ANOVA are presented in Table 4, indicating statistically significant differences in the main effects of muscle and vehicle speed as well as in the two-way interaction effects of muscle × vehicle motion and muscle × vehicle speed for the absolute correlation coefficient. However, there was no significant difference in the three-way interaction effect.

As illustrated in Figure 3, the main effect for muscle revealed a significantly high correlation coefficient between the TL, MD, BB, and SCM muscles with both groups A and B, as determined by the Tukey post hoc test.

Figure 4 shows a significant two-way interaction effect between the muscle and vehicle motion for the absolute correlation coefficient. For the neck part, the SCM muscle showed a significantly higher correlation with the LatAcc. For the shoulder part, the AD and MD muscles showed a significantly higher correlation with SWT, and the PD muscle showed a significantly higher correlation with SWA. In particular, the AD muscle showed a significantly higher correlation coefficient with SWT than with other vehicle motions. For the arm part, the BB muscle showed a significantly higher correlation with LatAcc, and the TL muscle showed a significantly higher correlation with SWA. The correlation coefficient between the facial and trunk parts and vehicle motions did not show a significant difference.

Figure 5 demonstrates a significant two-way interaction effect between the muscle and vehicle speed for the mean absolute correlation coefficient. The mean absolute correlation coefficient exhibited an increasing trend with the increasing vehicle speed for the muscles (increasing rate%): UT (30.5%), PMS (38.7%), SA (13.3%), and BB (11.0%). The mean absolute correlation coefficient showed a decreasing trend with increasing vehicle speed for the MA (−9.6%), SCM (−12.9%), MD (−5.5%), PD (−20.0%), PMC (−13.4%), and TL (−6.3%).

## 4. Discussion

Based on the results, the null hypothesis of this study was rejected, indicating that the muscular activities of certain muscles have a high correlation with vehicle motions and can effectively reflect the signals of vehicle dynamics. This study analyzed a significantly high correlation between the SCM muscle and vehicle motions, with a correlation coefficient above 0.6 (as indicated in Table 3). It means that SCM muscle was activated at turning right side of DLC driving (SWA was positive value). The lower EMG activity observed in the SCM muscle compared to other muscles might explain why the SCM muscle in the neck region was not directly associated with the steering wheel maneuvers of the driver. Nevertheless, this muscle showed a significantly high positive correlation with vehicle motions. Kempter et al. [34] reported that the kinematic behavior of the driver in stabilizing the head under an external motion platform is independent of head posture. Zheng et al. [22] reported that the EMG activity of the SCM significantly increased in response to lateral acceleration, indicating its role in stabilizing the head during slalom driving. Using a relevance vector machine model, Doshi and Trivedi [35] found that the true positive rate of a driver’s lane change intent increased significantly by approximately 30%. Interestingly, the SCM muscle contracts opposite to the direction of vehicle motions to stabilize the head against body shaking. This suggests that the head motion could potentially be used to predict the driver’s intention to control the steering wheel while driving. Because the neck rotates to the right with steering wheel control, the activity of the right SCM muscle is highly correlated with vehicle motions. Although this study did not consider muscles on the left side, they may be activated when turning left during DLC driving. As depicted in Figure 5, the mean absolute correlation coefficient for the SCM muscle decreased with increasing vehicle speed. Because the passing time through the DLC course decreased with increasing vehicle speed, the preliminary preparation time for the next steering wheel control was also reduced. This likely led to the decrease in the rotation angle of the neck. Consequently, the mean absolute correlation coefficient for the SCM muscle may decrease with increasing vehicle speed because the EMG activity of the SCM muscle decreases as neck rotation decreases. Based on these results, SCM muscle effectively reflects vehicle motions and should be considered important in evaluating vehicle performance.

As depicted in Figure 4, the AD muscle exhibited a significantly higher correlation with SWT in the shoulder part. The AD muscle serves as the primary flexor of the shoulder, which is necessary for flexion during steering wheel control in DLC driving [4,36]. During DLC driving, Pick and Cole [18] evaluated that a multiple regression model including AD and MD muscles demonstrated good predictive ability for vehicle motion related to steering wheel torque (SWT), as evidenced by a multiple correlation coefficient above 0.88. Figure 5 shows that the mean absolute correlation coefficient for the deltoid muscles decreased with increasing vehicle speed. This decrease may be attributed to the dramatic increase in the rolling resistance coefficient caused by high-energy standing waves when the vehicle speed exceeded 60 km/h [20,37]. The correlation between vehicle motion and the activity of shoulder stabilizer muscles, such as the UT, PMS, and SA muscles, increased. This is likely because the use of shoulder stabilizer muscles to stabilize the glenohumeral joint becomes activated when vehicle speed increases. Therefore, these shoulder muscles should be considered, as they are related to the control of the steering wheel and influence the vehicle motion for steering wheel torque during driving.

The absolute correlation coefficients between the UT, PMS, and SA muscles and the vehicle motions did not show a significant correlation. However, the trends in the absolute correlation coefficient consistently increased as the vehicle speed increased. The UT muscle is known to be sensitive to muscle reflective of stress and strain [38,39]. Lundberg et al. [40] reported that the EMG activity of the UT muscle increased by 20% in the mental arithmetic condition compared to the rest condition, indicating a positive correlation between EMG activity of the UT muscle and mental stress. Therefore, the increasing stress and strain experienced by the driver could lead to an increase in the correlation between the EMG activity of the UT muscle and the vehicle motion as the vehicle speed increases.

As the vehicle speed increases, the lateral acceleration tends to increase in the opposite direction of vehicle turning. In response, the driver must maintain his body balance to resist lateral acceleration. Indeed, as vehicle speed increases, more strength is required to maintain body balance and resist the effects of lateral acceleration [22]. Lee et al. [41] and Watanabe and Yoshida [42] reported that the upper extremity muscles are closely related to driving owing to the recruitment of various muscle fibers/motor units during the driver’s steering wheel maneuver. These results indicate that the tension of the UT muscle increases to maintain an upright upper body position. Despite the low correlation of UT, PMS, and SA muscles with vehicle motions, these muscles, which act as shoulder stabilizers, are considered to play a primary role in maintaining body balance while driving.

In the upper arm part, the correlation coefficients of the BB and TL muscles showed a significantly high correlation with vehicle motions, above 0.7, and the signs of correlation coefficients of these muscles showed an inverse relationship. This result may stem from the different functions of each muscle. The BB muscle, serving as an agonist of elbow flexion, was activated during sections of right-side turning (+SWA), while the TL muscle was activated during sections of left-side turning (−SWA). In other words, during right-side turning, the right elbow flexed while the left elbow extended, and vice versa during left turning. The absolute correlation coefficient of the BB muscle increased, but that of the TL muscle decreased as the vehicle speed increased. This is because the driver’s body balance shifted towards the lateral side owing to the increasing lateral acceleration at higher vehicle speeds. The sway of body balance necessitates more precise steering wheel control, leading to characteristics of EMG such as the duration of activation and on/off-set time being influenced by neural compensation of the central nervous system [34,43,44]. Deschamps et al. [45] reported that EMG activity and activated duration of BB increased and onset time was faster during repetition of flexion and extension of the elbow. These results suggest that the neural adaptation effect occurs to adapt to different conditions. The decreasing passing time through the DLC course owing to increasing vehicle speed demanded more precise steering wheel control. Consequently, EMG activity of the BB muscle increased, whereas that of the TL muscle decreased. Therefore, the BB and TL muscles should be considered as the main muscles that reflect the driver’s response while driving.

The FCR, ECR, FDS, FDP, and EDC muscles for the lower arm part exhibited low correlation coefficients with vehicle motions, all less than 0.3. Additionally, the variations in EMG activities of the lower arm muscles were lower than those of the shoulder and upper arm muscles. Therefore, this study concluded that the lower arm muscles did not significantly correlate with vehicle motion. For the abdominal part, the RA muscle also exhibited a low correlation with vehicle motion. This result can be attributed to the experimental conditions in this study, where DLC at a constant speed caused the rolling of the vehicle rather than pitching. The EMG activity of the RA muscle was low because lateral bending or rotation of the trunk occurred instead of flexion and extension of the trunk. To identify the muscles reflecting vehicle motion, further studies should consider the EMG activity of the abdominal external oblique muscle, which is related to the lateral bending or rotation of the trunk.

A limitation of this study was the small sample size of professional drivers, which may make it challenging to draw general conclusions. Further studies are needed to understand the pattern of muscle activity by including a larger and more diverse group of subjects, which should also encompass the evaluation of lower limb muscles. Additionally, it will be important to integrate various physiological and physical signals, such as muscular activity, heart rate variability, and body pressure on the seat, to assess driver-vehicle interactions during real driving.

## 5. Conclusions

This study evaluated the correlation between the EMG activity and vehicle motion during DLC driving. SCM, MD, PD, BB, and TL muscles exhibited a high correlation with vehicle motions. The AD muscle, which is primarily responsible for generating shoulder torque, showed a higher correlation with vehicle motion for the SWT. Despite UT, PMS, and SA muscles showing a low correlation with vehicle motions, these muscles, as shoulder stabilizers, are considered to play a primary role in maintaining body balance while driving. Therefore, this study suggests that the primary muscles reflecting vehicle motions include the SCM, AD, MD, PD, UT, PMS, SA, BB, and TL muscles under real driving conditions. The findings of this study are expected to be widely used in this study of driver-vehicle interaction. Accordingly, since muscular activity can effectively reflect vehicle motion, it is possible to assess both the driver’s muscle responses and vehicle motions without the need for installing heavy and potentially hazardous vehicle motion measurement equipment inside the vehicle. These findings contribute to a better understanding of muscular activities, which can be considered highly related to vehicle motion.

## Figures and Tables

**Figure 1 sensors-24-05982-f001:**
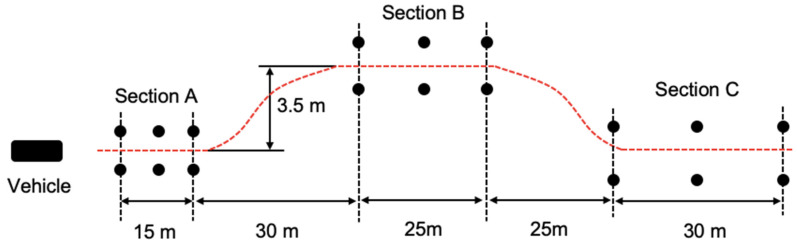
Double lane change course (vehicle width is 1.79 m).

**Figure 2 sensors-24-05982-f002:**
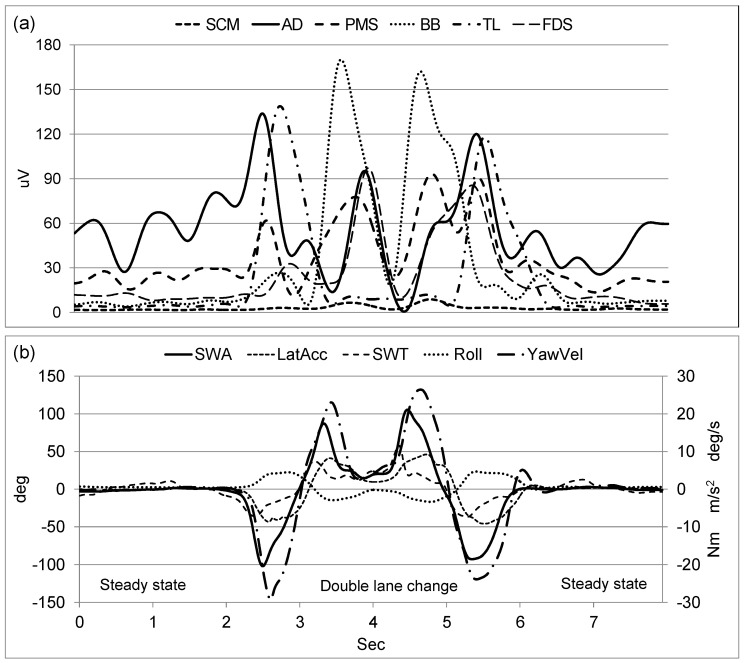
Example signals of EMG activities (**a**) for sternocleidomastoid (SCM), anterior deltoid (AD), pectoralis major sternal (PMS), biceps brachii (BB), triceps brachii long head (TL), and flexor digitorum superficialis (FDS) and vehicle motions (**b**) for steering wheel angle (SWA), steering wheel torque (SWT), lateral acceleration (LatAcc), roll angle (Roll), and yaw velocity (YawVel) during double lane change at 100 km/h speed.

**Figure 3 sensors-24-05982-f003:**
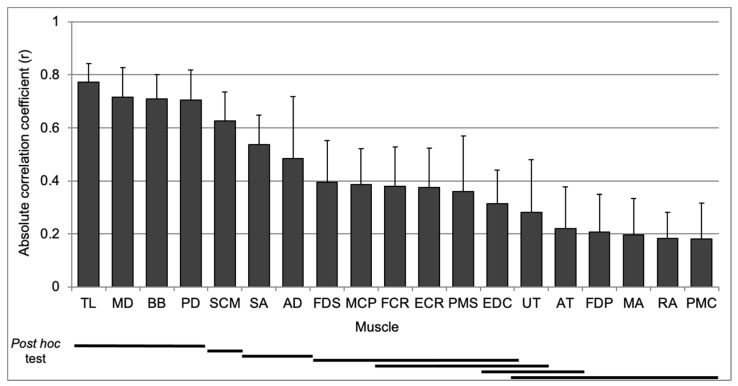
Mean absolute correlation coefficient for muscle main effect with Tukey post hoc test.

**Figure 4 sensors-24-05982-f004:**
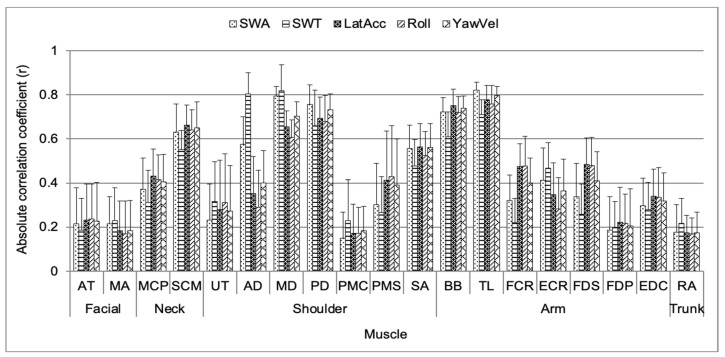
Mean absolute correlation coefficient for two-way interaction effect between muscle and vehicle motion.

**Figure 5 sensors-24-05982-f005:**
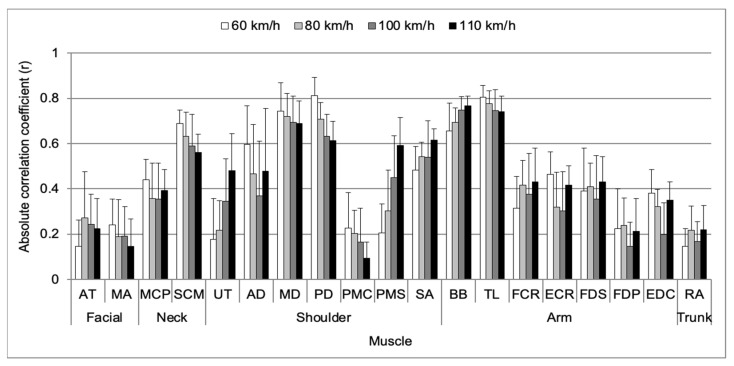
Mean absolute correlation coefficient for two-way interaction effect between muscle and vehicle speed.

**Table 1 sensors-24-05982-t001:** The description of the vehicle axis system and five vehicle motion variables.

Vehicle Motion (Abbreviation)	Unit	Description
Steering wheel angle (SWA)	deg	The angle between x axis of the vehicle and the wheel plane
Steering wheel torque (SWT)	Nm	The total torque about a tire’s steer axis resulting from the tire force
Lateral acceleration (LatAcc)	m/s^2^	The component of the vector acceleration of a point in the y axis
Roll angle (Roll)	deg	The rotation angle between the vehicle x axis and the ground plane
Yaw velocity (YawVel)	deg/s	The angular velocity between the z axis and the ground plane
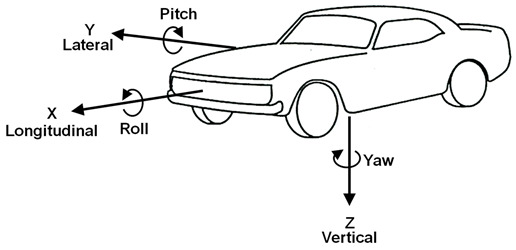		

**Table 2 sensors-24-05982-t002:** The position of 19 muscles using surface electrodes.

Body Part	Muscle (Abbreviation)	Electrode Position
Facial	Anterior temporalis (AT)	Placed at two-figures’ breadth above the zygomatic arch and two-fingers’ breadth posterior to the eye commissurae
	Masseter (MA)	Placed at one-finger’s breadth posterior to the anterior edge of the muscle and one-finger’s breadth cephalad to the lower edge of the mandible
Neck	Midcervical paraspinal C4 (MCP)	Placed at approximately 2 cm for the midline, over the muscle belly at approximately C-4
	Sternocleidomastoid (SCM)	Placed at four-fingers’ breath cephalad to the muscle origin (at the level of the thyroid cartilage)
Shoulder	Upper trapezius (UT)	Placed at 50% on the line from the acromion to the spine on vertebra C7
	Anterior deltoid (AD)	Placed at one finger width distal and anterior to the acromion
	Middle deltoid (MD)	Placed from the acromion to the lateral epicondyle of the elbow, which should correspond to the greatest bulge of the muscle
	Posterior deltoid (PD)	Placed at the center, the electrodes are in the area about two finger widths posterior to the acromion
	Pectoralis major clavicular (PMC)	Placed at approximately 2 cm below the clavicle, just medial to the axillary fold
	Pectoralis major sternal (PMS)	Placed at the anterior axillary fold
	Serratus anterior (SA)	Placed at just lateral to inferior angle of scapula
Arm	Biceps brachii (BB)	Placed on the line between the medial acromion and the fossa cubit at 1/3 from the fossa cubit
	Triceps brachii long head (TL)	Placed at 50% on the line between the posterior crista of the acromion and the olecranon at two finger widths medial to the line
	Flexor carpi radialis (FCR)	Placed four fingerbreadths distal to the midpoint of a line connecting the medial epicondyle and biceps tendon
	Extensor carpi radialis (ECR)	Placed at two fingerbreadths distal to lateral epicondyle
	Flexor digitorum superficialis (FDS)	Placed at index finger to biceps tendon and insert needle electrode just ulnarly to tip of index finger
	Flexor digitorum profundus (FDP)	Placed at tip of little finger on olecranon and ring, middle and index finger along shaft of ulna
	Extensor digitorum communis (EDC)	Placed at index finger bisect two points and attach electrode at tip of index finger to a depth of one-half inch
Trunk	Rectus abdominis (RA)	Placed at two-finger breadths lateral to the abdominal midline

**Table 3 sensors-24-05982-t003:** The mean correlation coefficients (SD) between the muscle and vehicle motion (for easy viewing, the high correlation coefficient (r > 0.6) is shown in highlights).

	SWA	SWT	LatAcc	Roll	YawVel
AT	0.19 (0.08)	0.17 (0.11)	0.21 (0.08)	−0.22 (0.07)	0.20 (0.07)
MA	−0.03 (0.09)	0.06 (0.11)	−0.06 (0.07)	0.08 (0.06)	−0.06 (0.07)
MCP	0.36 (0.06)	0.31 (0.06)	0.43 (0.03)	−0.42 (0.04)	0.40 (0.04)
SCM	0.61 (0.08)	0.54 (0.05)	0.67 (0.04)	−0.64 (0.02)	0.64 (0.07)
UT	0.14 (0.21)	−0.11 (0.28)	0.28 (0.24)	−0.32 (0.22)	0.27 (0.20)
AD	−0.58 (0.07)	−0.79 (0.06)	−0.37 (0.13)	0.25 (0.14)	−0.40 (0.08)
MD	−0.79 (0.01)	−0.80 (0.11)	−0.68 (0.04)	0.60 (0.04)	−0.71 (0.03)
PD	−0.74 (0.08)	−0.64 (0.15)	−0.70 (0.08)	0.67 (0.08)	−0.73 (0.04)
PMC	−0.03 (0.06)	−0.18 (0.14)	0.07 (0.07)	−0.11 (0.07)	0.06 (0.12)
PMS	0.27 (0.21)	0.05 (0.29)	0.41 (0.24)	−0.44 (0.23)	0.40 (0.19)
SA	0.56 (0.04)	0.49 (0.07)	0.57 (0.04)	−0.53 (0.05)	0.57 (0.03)
BB	0.73 (0.03)	0.62 (0.09)	0.76 (0.05)	−0.73 (0.05)	0.74 (0.02)
TL	−0.82 (0.02)	−0.70 (0.05)	−0.78 (0.03)	0.75 (0.04)	−0.80 (0.01)
FCR	0.31 (0.05)	0.20 (0.09)	0.46 (0.08)	−0.49 (0.10)	0.40 (0.04)
ECR	0.41 (0.07)	0.47 (0.06)	0.36 (0.08)	−0.28 (0.07)	0.37 (0.07)
FDS	0.33 (0.04)	0.24 (0.09)	0.46 (0.05)	−0.49 (0.06)	0.40 (0.03)
FDP	0.15 (0.07)	0.20 (0.09)	0.16 (0.08)	−0.13 (0.09)	0.16 (0.08)
EDC	0.29 (0.08)	0.29 (0.09	0.33 (0.08)	−0.33 (0.09)	0.31 (0.08)
RA	−0.04 (0.13)	−0.03 (0.19)	−0.06 (0.07)	0.06 (0.03)	−0.04 (0.09)

**Table 4 sensors-24-05982-t004:** The result of ANOVA for absolute correlation coefficient between EMG activity and vehicle motion.

Source	df	MS	*F*	Pr *> F*
Muscle	18	11,565,018.0	219.72	<0.0001
Vehicle motion	4	120,479.6	2.29	0.0579
Vehicle speed	3	414,653.6	7.88	<0.0001
Muscle × Vehicle motion	72	315,135.9	5.99	<0.0001
Muscle × Vehicle speed	54	392,718.1	7.46	<0.0001
Vehicle motion × Vehicle speed	12	41,249.6	0.78	0.6679
Muscle × Vehicle motion × Vehicle speed	216	45,787.0	0.87	0.9012
Error	1220			
Total	1599			

## Data Availability

The data used during the current study are available by the authors on request.

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
