# Peer review of "Correlation between Muscular Activity and Vehicle Motion during Double Lane Change Driving"

_sensors, 2024, doi:10.3390/s24185982_

Round 1
Reviewer 1 Report
Comments and Suggestions for Authors
The authors submitted a paper on the correlation between muscular activity and vehicle motion during double lane change. Proposed study is important in explaining the physiology of driving a car and improving road safety.
-The abstract is very short and does not contain the most important results, i.e. the correlation coefficient between the muscle activity and vehicle movement. Moreover, in line 17, I would add "the activity of the" before the names of the muscles.
Study design:
-What type of car was used?
-Why did you decided not to carry out the study on higher numbers of subjects?
-What are the steps of the pattern recognition function? Was it described below in lines 191-197?
Minor points:
Line 99: Dusseldorf → Düsseldorf
Line 107: got familiar → were familiarized
Line 117: NORAXON, USA → Noraxon USA, Inc., Scottsdale, AZ, USA
Line 119: All the data → All data
Line 132: See the comment for line 117
Line 171: "for the ISO; 3888-1" → "for the ISO 3888-1"
Line 190: See the comment for line 117
Lines 196-197: 7.0.4, The Mathworks Inc. → version 7.0.4, MathWorks, Inc., Natick, MA, USA
Figures: I would consider using colours to distinguish different variables, not only line styles.
Tables: The tables (except for Table 2) are split between the pages (the caption and the header remain on the page A and the rest is moved to page B). Table 1 is long but this is not a surprise.
Comments on the Quality of English Language
The quality of English is good enough but to improve clarity, I would consider minor proofreading.
Author Response
Thank you very much for taking the time to review this manuscript. Please find the detailed responses below and the corresponding revisions highlighted in track changes in the re-submitted files.

Reviewer 2 Report
Comments and Suggestions for Authors
Studies highlight significant correlations between electromyography activity in various muscles and key vehicle dynamics, such as steering angle and lateral acceleration. Below, I address some questions regarding the manuscript:
1.Abstract: Include the type of study conducted.
2.Abstract: Insert quantitative data to provide a clearer view of the magnitude of the identified correlations.
3.Introduction: What gaps in the literature does this study aim to fill?
4.Introduction: The introduction mentions previous studies; could you include a direct comparison with other relevant studies that also explored the relationship between electromyography and vehicle movement? How do the results of this study differ from previous ones?
5.Introduction: Add the study’s null hypothesis.
6.Introduction: Are there more recent studies that could be included to strengthen the theoretical foundation?
7.Materials and Methods: Mention the possibility of bias in data collection that was not addressed in the study.
8.Materials and Methods: Provide more information about the selection of study participants. What criteria were used to determine driver suitability (e.g., experience, age, health)?
9.Materials and Methods: Did the study data present a normal distribution?
10.Materials and Methods: Include details about the calibration and accuracy of the equipment used.
11.Discussion: Was the null hypothesis accepted or rejected?
12.Discussion: Could the authors elaborate on how the findings might influence vehicle design or driver safety?
13.Discussion: Are there other limitations that should be discussed?
14.Discussion: The manuscript presents suggestions for future research. Be more specific about which aspects should be explored in future studies.
Author Response

(The authors gave the same response as above.)

Round 2
Reviewer 2 Report
Comments and Suggestions for Authors
The authors responded to the questions.